# Identification of Plastic Type and Surface Roughness of Film-Type Plastics in Water Using Kramers–Kronig Analysis

**Boniphace Elphace Kanyathare [1,2,\*], Benjamin Asamoah [1] , Muhammad Umair Ishaq [1], James Amoani [1] , Jukka Räty [3] and Kai-Erik Peiponen [1,\*]**

[1]  Department of Physics and Mathematics, University of Eastern Finland, FI-80100 Joensuu, Finland; benjamin.asamoah@uef.fi (B.A.); shiekhumair99@gmail.com (M.U.I.); jamessamoani@gmail.com (J.A.)

[2]  Department of Electronics and Telecommunication Engineering, Dar Es Salaam Institute of Technology, 2958 Dar Es Salaam, Tanzania

[3]  MITY, University of Oulu, Technology Park, FI-87400 Kajaani, Finland; jukka.raty@oulu.fi

[\*]  Correspondence: boniphace.magina@dit.ac.tz (B.E.K.); kai.peiponen@uef.fi (K.-E.P.)

**Abstract:** The knowledge of the plastic type, thickness, and the nature of the surface is important towards the monitoring of microplastic pollution in water bodies, especially when vis-NIR spectroscopy is utilized. Factors such as complex environment and surface roughness induced-light scattering of the probing light limit the optical detection of these parameters in in-situ measurements, however. In this paper, a novel application of Kramers–Kronig analysis was exploited to identify both smooth and rough film-type macroplastics with unknown thickness. This method is particularly useful in the in-situ identification of unknown film-like macroplastics; although the sample is large, the ratio function is detected from an area that corresponds to the size of a MP. Therefore, it can be applied for the case of large size MPs. The validity of the method was demonstrated using transmittance data for smooth and roughened plastics given in Kanyathare et al., 2020.

**Keywords:** microplastic; Kramers–Kronig relation; surface roughness; transmittance; vis-NIR spectroscopy

## 1. Introduction

Microplastic (MP) pollution of water bodies enables, for example, the growth and transfer of bacteria, especially when the MPs are rough, rendering it a substrate for toxic contaminants [1]. Sources of microplastic pollution include pristine transparent plastic films commonly used for food preservation and packaging. Unlike other plastic types, plastic films can decompose into film-type MPs, namely macroplastics, that appear both in natural aquatic environments and municipal wastewaters. In water, the colorless and translucent plastic films turn transparent. On the other hand, colored plastic films can fade and become translucent in water, complicating the method of detection.

Microplastics pollution endangers the health of humans and water-inhabitant species [1,2]. In addition to organism-related issues, microplastic pollution also threatens water quality, requiring an urgent need to develop techniques for the in-situ detection of MPs in natural water bodies and municipal water treatment systems.

Several optical methods have been developed for the detection of plastics. For example, remote sensing of floating microplastics by airborne hyperspectral imaging has been demonstrated [3]. However, in the case of microplastic detection embedded in water, the same method will require further development. Typically, to detect these microplastics, they are harvested from aquatic environments and measured as dry samples to avoid the attenuation of radiation. This technique is performed to limit water absorption [4] in the near infrared (NIR) spectral range. However, the existence of different

second and third-order overtones for hydrocarbons of different plastic types in the vicinity of 1200 nm [3–5], where water absorption is not so strong, can be exploited for plastic identification. Despite the challenge of water absorption, Vázquez-Guardado et al. used a multispectral infrared spectroscopic technique to successfully classify 12 plastic resins found in municipal waste by considering plastics in the air [6]. Peiponen et al. have also demonstrated the possibility to detect and identify smooth plastics embedded in water by using the transmission spectrum in the NIR region, making use of the known sample thickness [7]. Recently, other interesting techniques, such as surface-enhanced Raman spectroscopy [8], laser-based photoluminescence [9], and integrated holography and Raman spectroscopy [10], have been proposed for the detection of microplastics. These methods exploit the spectral properties of plastics in their detection.

In addition to the spectral properties of the plastics, physical properties, such as the size and surface texture, which are dependent on mechanical and chemical erosion in aquatic environments, also influence the measured signal. The erosion causes, for example, surface roughening of the microplastic [11]. Surface roughness is an indication of the aging of the microplastic and also serves as a substrate for bacterial and virus growth [12–14] and contamination by toxic materials such as metals [2] and antibiotics [15]. Hence, it is important to obtain information on the surface roughness of microplastics. Recently, we have suggested optical sensing techniques to monitor colorless flat or curved MPs with surface roughness in water [16–18]. In [18], we used a priori known thickness of plastics for monitoring of the surface roughness of the plastics. However, for field measurement conditions, both the thickness and the plastic type are unknown and an analytical method like Kramers–Kronig (K-K) is better suited for such a problem.

In this work, we apply a non-conventional K-K relation [19] on inverting transmittance for the identification of smooth (pristine) and rough film-type macroplastics with unknown thickness. The macroplastics were prepared from common plastics that are commonly found in wastewater [20]. Despite the dominance of secondary microplastics of these plastics with common morphologies such as fiber, particles, and foam, film-type macroplastics pollution is also emerging [21]. We, therefore, consider such film-type macroplastics which can be easily probed with a spectrophotometer at an area that corresponds to the size of a MP. Henceforth, we refer to all the film-type macroplastics in this study as macroplastics for simplicity. Using the K-K relation and the measured transmittance of the macroplastics, we respectively calculate the change in the optical path and the attenuation lengths; and determine the ratio (curves) of the two parameters. We validate the method of ratio curves using the transmittance data of smooth and rough macroplastics presented in [18].

The key point is to exploit a library of ratio curves obtained from the intrinsic optical properties of the plastics for the identification of plastic type and the determination of their surface roughness in the aquatic environment. A similar concept of the ratio curve was presented in [22] for a non-scattering medium, however, using the product of sample thickness and complex refractive index change. Here, the novel feature in the application of the ratio curves is that the sample thickness is irrelevant in the frame of macroplastics. Additionally, the ratio curves are applied to screen both non-scattering and diffuse samples. The diffuse samples scatter incident vis-NIR radiation due to the presence of surface roughness. We further remark that the K-K analysis, although in a different form than what is presented in this study, has been previously applied in detecting concealed diffuse objects [23]. This study is useful towards the developing of optical sensors for the identification of smooth and rough MPs in wastewater and in other aquatic environments.

## 2. Theory

When light interacts with media such as microplastic, several phenomena such as reflection, refraction, absorption and transmission might be involved. The strength or weakness of these phenomena depends on what is known as the wavelength-dependent complex refractive index of the medium. The complex refractive index is expressed as $N = n + ik$, where $n$ and $k$ denote the real refractive index and the so-called absorption index of the medium, respectively. The imaginary unit, $i$, is defined as

$\sqrt{(-1)}$. The complex refractive index is an intrinsic property of a medium and, therefore, holds both for medium having either a smooth or rough surface.

The inversion of transmittance data to obtain the real refractive index of an insulator is straightforward for amorphous isotropic media. In the case of birefringent media, such as certain plastics, for example, polystyrene (PS) under mechanical tension, the non-conventional K-K relation is valid [24]. The samples used in this study are not birefringent; hence, we exploit a non-conventional K-K relation to determine the change in refractive index (optical path length). Here, we consider the case where the transmission or absorption information is available, but the sample thickness is unknown. This situation is typical in spectroscopic and noncontact measurements such as in this study.

We apply the generalized Beer-Lambert intensity law, where the absorption coefficient may contain both intrinsic absorption and scattering coefficients, to the measured transmittance of the samples. For non-scattering samples, Beer-Lambert's law is given by Equation (1). We emphasize that the use of the Beer-Lambert law is valid for an ideally flat film and intrinsically assumes the detection to be within the sample. However, both assumptions cannot be satisfied in practical situations. In the case of macroplastic samples with thickness nearly comparable to the wavelength of the probing light, one would observe periodic oscillations in the transmittance curve due to interference. The appearance of such interference fringes would give information on the thickness of the plastic film. However, such interference was absent. The method described in this study presents a useful tool in the analysis of in situ optical detection of macroplastics and other larger microplastics in water.

$$I\,(\lambda) = I_0(\lambda)exp[-\alpha(\lambda)L \tag{1}$$

where $I_0$ is the incident intensity, $I$ the transmitted intensity, $\alpha$ is the absorption coefficient, $\lambda$ is the wavelength of the probe light, and $L$ is the thickness of the sample. The relation in Equation (1) is formal and neglects the scattering of electromagnetic radiation. The absorption coefficient, $\alpha$, is linked to the extinction coefficient by $k = c\alpha/(2\omega)$, where c is the speed of light in the vacuum and $\omega$ is the angular frequency of the probe light. If the experimental data on the transmittance, $T_i(\lambda) = I(\lambda)/I_0(\lambda)$, are available, one can obtain the product $kL = -c/(2\omega)lnT_i(\lambda)$ without knowing the sample thickness a priori; it is possible to calculate the inversion of $kL$ data using the non-conventional K-K relation. The K-K relation is valid for non-scattering samples and it is given by [22] as:

$$\Delta n(\omega')L = (n(\omega') - 1)L = \frac{2}{\pi}P \int_0^\infty \frac{\omega k(\omega)L d\omega}{\omega^2 - \omega'^2} \tag{2}$$

where $\Delta n(w') = n(w') - 1$, $\Delta n$ denotes the frequency-dependent refractive index change and $P$ is the Cauchy principal value. Since the non-conventional K-K relation implicitly neglects the scattering of EM radiation of the probe light, samples with volume or surface roughness will scatter incident radiation, resulting in the difference in the calculated values of $\Delta n(\omega')L$ ($\Delta nL$) for smooth and rough samples. It is, therefore, possible to form the ratio ($\sigma$) between the measured and calculated values of ($kL$) and ($\Delta nL$) as:

$$\sigma = kL/\Delta nL \tag{3}$$

where $kL$ is the attenuation length and $\Delta nL$ is the change in the optical path length. This ratio is useful in many ways. As is evident, the thickness of the sample becomes irrelevant. The ratio curves can also be used for the identification of plastic types especially in the vicinity of 1200 nm, where there exists different 2nd and 3rd overtones for different plastic types. Moreover, working in the vicinity of 1200 nm, one avoids strong water absorption in the vicinity of 1500 nm, which restricts the transmittance measurement over a long detection distance. Additionally, the ratio also compensates for the temperature dependence of $kL$ and $\Delta nL$. This compensation comes from the fact that any change in $kL$ results in a proportional change in $\Delta nL$, which is a typical feature in the absorption–dispersion behavior of light in a medium. This behavior is accurately predicted by the most general dispersion relation of light, namely the K-K relation. The cancellation of the effect of temperature change in the

ratio curve is another practical advantage in designing a spectrophotometer and analyzing its data. In principle, such an inexpensive spectrophotometer [25] can be economical for the monitoring of environmental pollution, such as macro and microplastics in aquatic environments.

## 3. Experiment

In this work, we have studied film samples having both faces smooth (smooth) and film samples having both faces rough (rough) with a size bigger than a microplastic. The samples are commercial plastics: polypropylene (PP), polyethylene terephthalate (PET), polyamide nylon 6 (PA), polystyrene (PS), and low-density polyethylene (LDPE), which were purchased from Goodfellow, Cambridge, UK. Thin rectangular sheets with an approximate size of $55 \times 7$ mm were cut from the large pristine plastic samples to represent the smooth macroplastics. Next, a larger piece of the pristine sample was roughened on both sides using different sandpaper grits in a circular pattern. A similar size to that of the smooth macroplastics was then cut from the roughened larger piece to represent the large macroplastics. The roughening was performed to simulate mechanical erosion of plastics in natural waterbodies. Tables 1 and 2 respectively show the thickness of the samples as purchased and the obtained average surface roughness, measured with a stylus profilometer (Mitutoyo SJ-210, Sakado, Japan), for the different sandpaper grits after roughening. The thickness of PET plastic is comparable to the thickness of plastic mineral water bottles. We, therefore, consider the special case of PET bottles having the same average surface roughness on both sides for different magnitudes to simulate of wear of plastics in water bodies.

**Table 1.** Plastic materials and thickness as purchased.

| Sample | Thickness (mm) |
|:------:|:--------------:|
| PP | 0.55 |
| PET | 0.25 |
| PA | 0.50 |
| PS | 0.19 |
| LDPE | 0.50 |

**Table 2.** Sandpaper grit size and the measured average surface roughness of the samples.

| Samples | Grit | Average Surface Roughness Ra (µm) |
|:-------:|:----:|:---------------------------------:|
|  | 1200 | $0.34 \pm 0.04$ |
| Rough | 320 | $1.10 \pm 0.17$ |
| Pristine | - | <0.10 |

The smooth and rough macroplastics were immersed in a 1mm-thick cuvette, containing tap water, to form a complex interface (air–glass–water–plastic–water–glass–air), and was measured with a spectrophotometer (Perkin-Elmer Lambda 9) in the vis–NIR spectral range under standard laboratory conditions. The smaller cuvette was chosen to reduce the absorption in the measured spectral range. The recorded transmission spectra for the different samples are available in [18].

We remark that in Figure 4 in [18], it is shown that the sensitivity of the transmittance against average surface roughness (at the spectral range starting from 500 to 1500 nm) is rather high because it can screen between a sample having average surface roughness magnitude 0.34 µm and a smooth plastic sample. Samples having higher magnitude of surface microroughness could be screened as shown in Figure 4 of [18].

## 4. Results and Discussion

The transmission spectra of both smooth and rough LDPE samples are shown as an example in Figure 1. The rough macroplastics clearly show the lowest transmittance than that of smooth macroplastics and water over the measured spectra range, except at about 1450 nm. The fingerprints

that are present for smooth plastic, in the vicinity of 1200, 1730, and 1760 nm, are also preserved in the case of the rough macroplastics, however. We also observe the dominant absorption band of water at 1450 nm in all three curves. The transmission spectra for the smooth and rough sample of the other macroplastics types, namely PA, PP, PS and PET, show similar characteristics; see [18]. These spectra were used for calculating the attenuation length (*kL*) as presented in Figure 2.

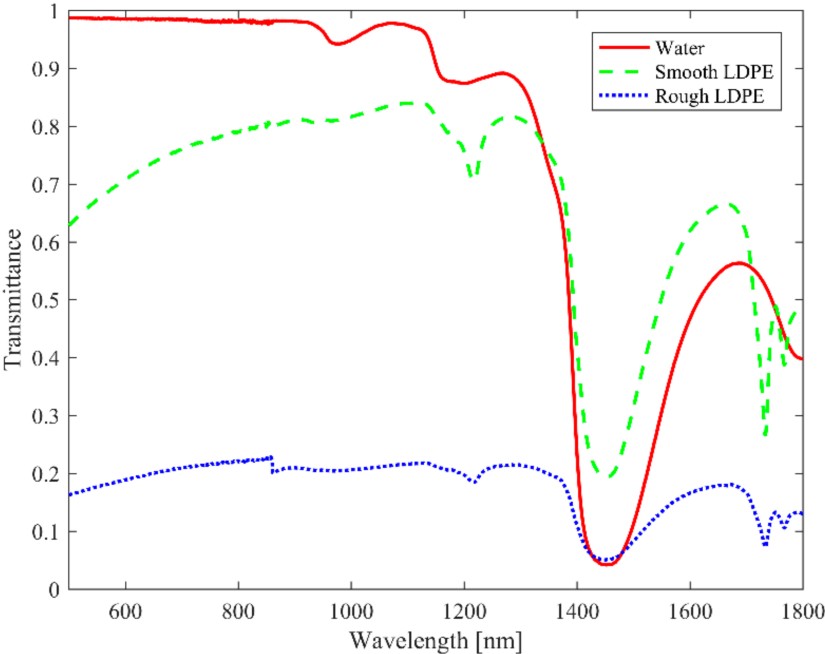

**Figure 1.** Vis–NIR transmittance spectra for water, smooth and rough LDPE macroplastics immersed in water volume.

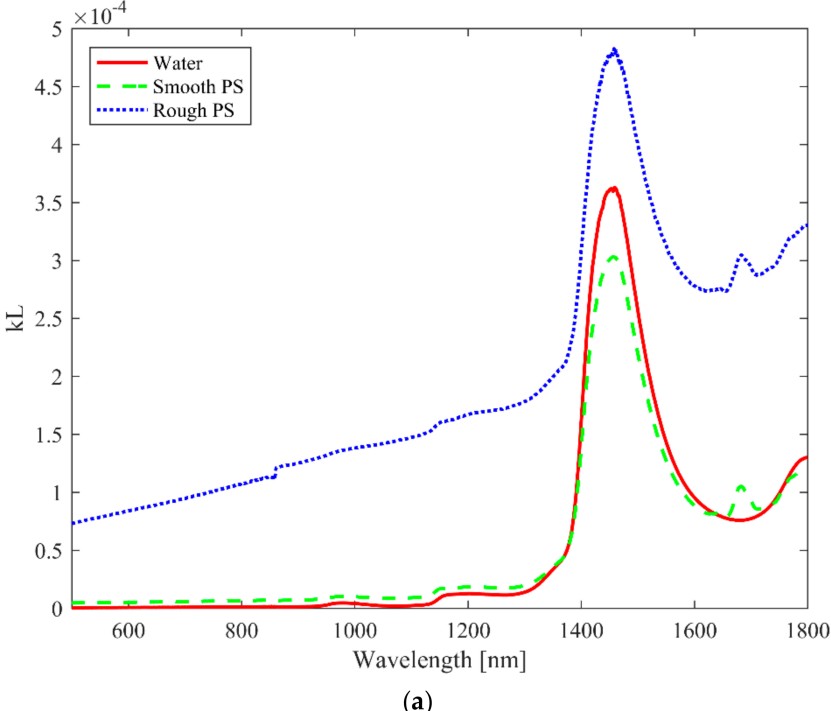

(**a**)

**Figure 2.** *Cont.*

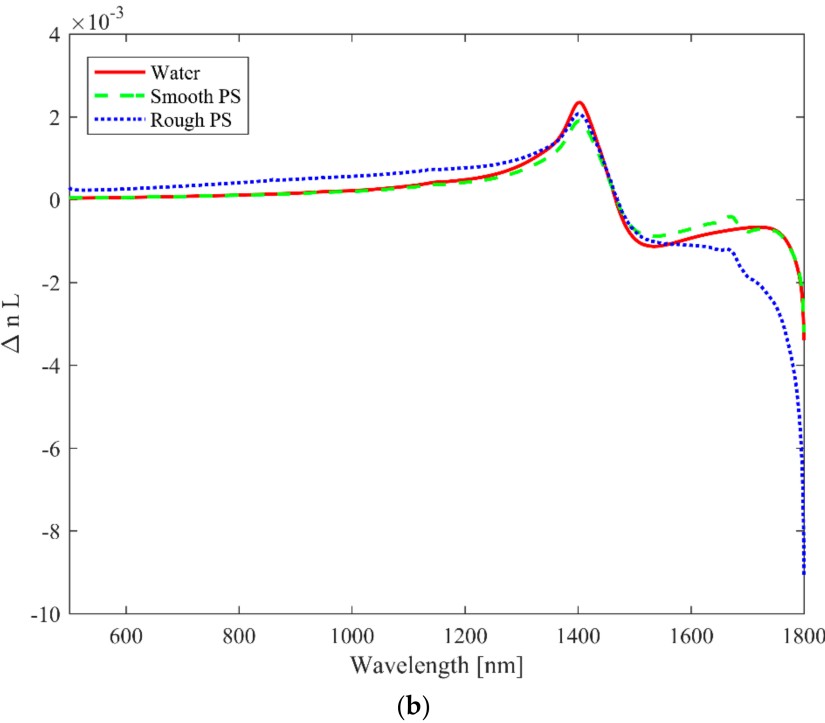

**(b)**

**Figure 2.** Product of imaginary/real refractive index for smooth and rough macroplastics, and water. (**a**) attenuation length (*kL*); (**b**) Change in optical path length (Δ*nL*).

As an example, Figure 2a shows the attenuation length (*kL*) curves for water and smooth and rough PA macroplastics embedded in water. The curves for water and the smooth macroplastics overlap at many points, while the curve for rough macroplastics is greatly separated from the other two. The wide separation of the *KL* curve for the rough macroplastics can be attributed to increased scattering or reabsorption. The separation of the *KL* curves from each other alone, in Figure 2a, is not sufficient for the differentiation of the smooth macroplastics from the rough as well as for the identification of the presence of macroplastics in water. Figure 2b shows the corresponding Δ*nL* curves calculated for all the samples using the MATLAB code available in [19]. Here, the curves are more tightly packed than in Figure 2a, which further complicates the discrimination and the identification of the macroplastics. We note that the curve in Figure 2b can be distorted in the vicinity of the initial and final wavelength due to the finite integration range of the measurement. To avoid any artificial data extrapolation beyond the range of measurement, such data points were excluded.

The data in Figure 2a,b were utilized to calculate the ratio σ for water, smooth, and rough macroplastics for the different plastic types in the spectral range of 900 to 1400 nm. All the smooth and rough macroplastics for the different plastics show higher ratio curves than water.

Figure 3a shows the ratio curves for PA macroplastics (rough and smooth) and water. The ratio curves for the smooth and rough macroplastics are well separated and can be differentiated from each other in the chosen spectral range. The spectral range was chosen to avoid the high absorption of light by both the macroplastics and water beyond 1400 nm. The ratio curve for water has a broad and very weak peak around 1200 nm, unlike in the normal transmittance curve, where the peak is dominant. The curve for smooth macroplastics shows a faint peak at 1161 nm but a stronger peak at 1211 nm, while the curve for rough macroplastics also shows similar peaks, but at 1166 and 1212 nm, respectively. We note the wavelength at which the smooth and rough macroplastics ratio curves intersect is likely to vary with changing average surface roughness. This conclusion is based on the fact the local transmittance is expected to increase when the average surface roughness corresponds to the wavelength in accordance with Mie scattering.

Figure 3b shows the ratio curves for the PP sample. The curves for the smooth and rough macroplastics intersect at 1233 nm. The smooth macroplastics shows three distinctive peaks at 1158,

1199, and 1222 nm that are absent in the curve for the rough macroplastics. Unlike in the case of the PA macroplastics, the trend of the ratio curve is relatively constant for the rough PP macroplastics over the spectral range considered, whereas that of the smooth decreases with increasing wavelength. Both rough and smooth LDPE macroplastics, in Figure 3c, show a relatively constant trend in their ratio curves with similar strong and weak peaks but no intersection points. The strong peaks are located at 1221 and 1224 nm, whereas the weak and broad peaks are located at 1171 and 1173 nm, respectively, for the rough and smooth macroplastics. The LDPE sample, unlike the other (PA, PP, PET, and PS) samples, already has volume inhomogeneity even for the pristine sample.

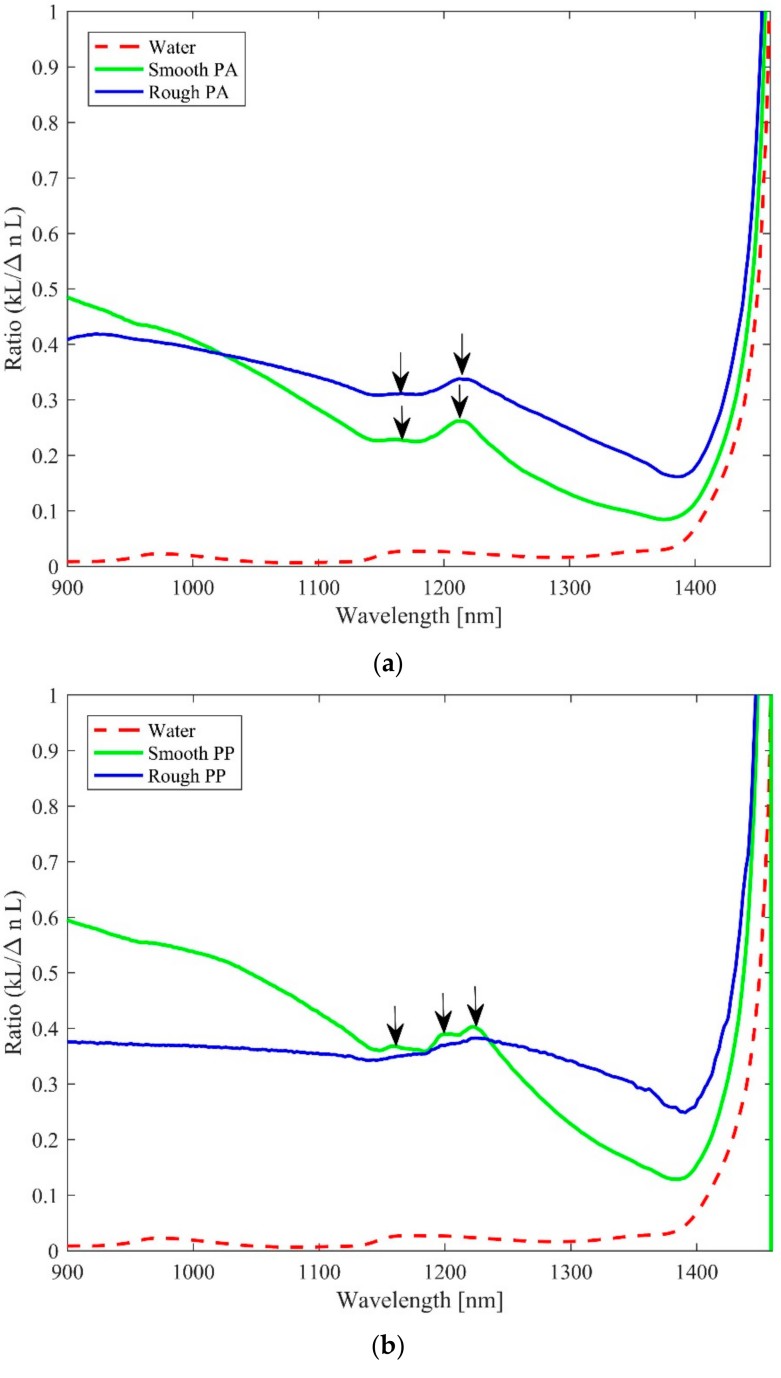

(a)

(b)

**Figure 3.** *Cont.*

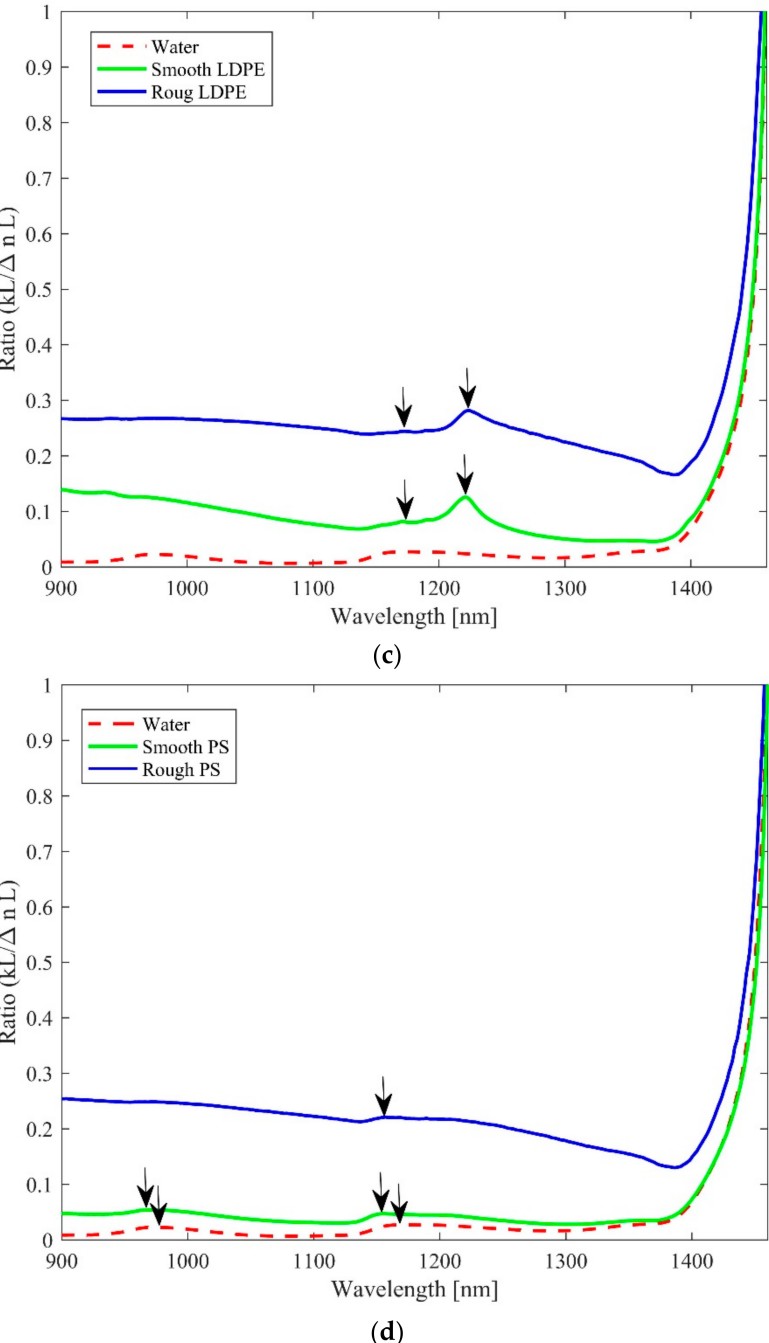

**Figure 3.** Ratio ($\sigma$) curves for smooth and rough macroplastics embedded in water. The short arrows indicate the peaks (**a**) PA type; (**b**) PP type; (**c**) LDPE type; (**d**) PS type.

Figure 3d also shows similar ratio curves for the PS sample. In contrast to the rough macroplastics that show a higher magnitude, the ratio curve for the smooth PS macroplastics is barely separated from that of water. The smooth PS, on the other hand, has two broad peaks at 969 and 1156 nm, whereas the rough PS macroplastic shows a single peak at 1157 nm. Besides these differences, both samples exhibit a similar relatively constant trend in their curves with no intersection point. In conclusion, the presence of macroplastics, either rough or smooth, in water can be detected based on the separation of their ratio curves from that of water. Additionally, rough and smooth macroplastics can be also be differentiated from each other based on the appearance and disappearance of the peaks, the intersection point as well as the trends in their ratio curves depending on the plastic type, as shown in Figure 3a–d.

Now, we examine how the ratio curve can be used for the discrimination of the different plastic types. Figure 4a,b illustrate the case for the smooth and rough macroplastics in water, respectively. The ratio curve for water shows the lowest magnitude in both figures as previously indicated. This observation suggests, again, that the ratio curve can be used for the identification of the macroplastics and other relatively large microplastics in water. The smooth macroplastics, in Figure 4a, show well-separated ratio curves for all the plastic types except that of PET and PS, which closely overlap. This overlap is because of the close similarities in the refractive index of the two plastics [7] which also manifest in the normal transmission spectra [18].

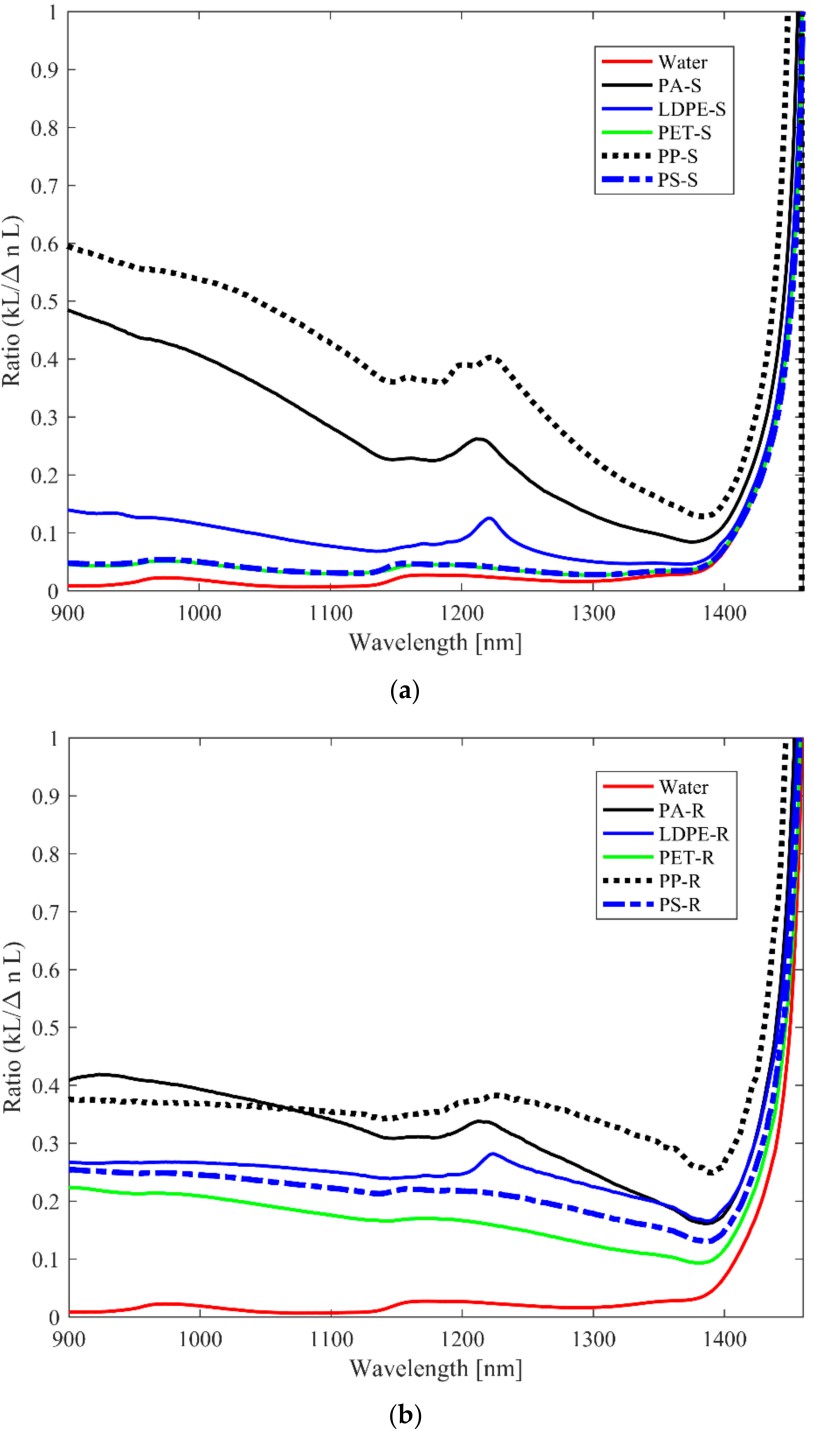

**Figure 4.** Ratio (σ) curves for macroplastics of the different plastic types (**a**) Smooth plastics; (**b**) Rough plastics.

The two overlapping curves are, however, well resolved. Moreover, we observe that the ratio curves in the case of the rough macroplastics, in Figure 4b, for all plastic types, is separated from that of water and the trend somewhat deviates from that of the smooth samples revealing the effect of the surface roughness. These observations can be exploited for the identification of macroplastics in water as well as the differentiation between rough and smooth macroplastics. The deviation in the ratio curves for the rough macroplastics can be attributed to the formation of a complex effective medium between the rough surface of the macroplastics and surrounding water [17].

A library of the ratio curves, therefore, presents a useful tool for the differentiation of (meso)plastics, different plastic types in the vis–NIR range, and to ascertain information on the surface roughness.

Lastly, we evaluate the suitability of the ratio curves for the discrimination of macroplastics with different average surface roughness, as illustrated in Figure 5. Only the PET plastic type is considered in this case for two average surface roughness, namely $Ra = 1.10$ μm and $Ra = 0.34$ μm. We observe a positive correlation of the magnitude of the ratio (curves) with increasing magnitude of average surface roughness of the plastics, indicating the sensitivity of former to the latter. As previously mentioned, the surface roughness creates a virtual layer at the top surface of the macroplastics as water infiltrates the grooves. This virtual layer has an effective index which corresponds to the ratio of the refractive indices of the PET plastic and water. The effect of such a layer on the reflection signal was examined in [17], however, using a more relaxed form of the effective medium theory [26]. For a relatively thin layer, a negative correlation was observed between the reflection signal and the magnitude of the average surface roughness, as one would expect, which agrees with the order of the ratio curves in Figure 5. In addition to the correlation and the separation of the curves, from that of water, with the increasing magnitude of $Ra$, we also observe weak peaks in the curves. There are two peaks for each of these curves in the vicinity of 960 and 1170 nm. Water has peaks at 973 and 1174 nm, and smooth macroplastics have peaks at 967 and 1174 nm. The PET-320 ($Ra = 1.10$ μm) has its peaks at 967 and 1168 nm, whereas PET-1200 ($Ra = 0.34$ μm) has one peak at 967 nm and the other shifted to 1170 nm.

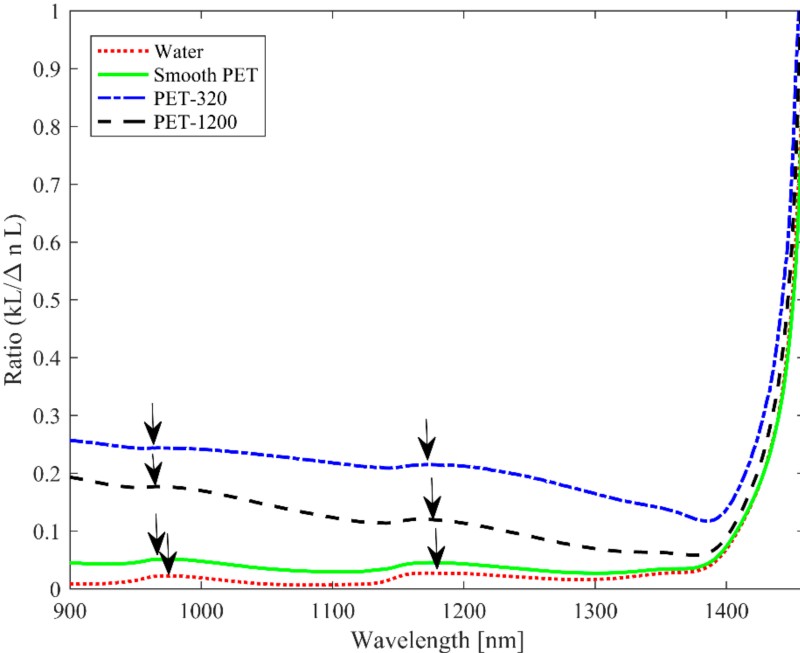

**Figure 5.** Ratio (σ) curves for water, smooth, and rough PET macroplastics with different average surface roughness water. The average surface roughness of the PET-320 and PET-1200 samples is Ra = 1.10 μm and Ra = 0.34 μm, respectively. The short arrows indicate the positions of the peaks.

The results are like those of transmittance in [18]; the ratio curve shows sensitivity to different average surface roughness, resulting in an increase in the magnitude of ratio and/or shift in peak

position. Such a sensitivity can be used for the differentiation among different macroplastics, with surface roughness on either or both surfaces. Moreover, the exponential behavior of transmittance versus roughness was observed in [27] at a fixed wavelength. This can be used to estimate the sensitivity of the present method versus average surface roughness, because in the first place prior to the Kramers-Kronig analysis, it is the transmittance that is measured. Sensitivity to find a change of at least of 300 nm of average roughness is reasonable.

In summary, it is difficult to use the individual parameters of the ratio ($\sigma$), namely the attenuation length ($kL$) and the change in optical length ($\Delta nL$), to differentiate among the plastics and also from that of water, especially for smooth samples as seen in Figure 2. On the contrary, for most of the plastic types, as seen Figure 3, the ratio curves follow a specific order with the curve for rough macroplastics being higher than that of the smooth macroplastics and the curve for water being the lowest. Moreover, the ratio (curve) enhances any spectral fingerprints in the vicinity of 1200 nm, which are suppressed in the case of the individual parameters due to the presence of strong absorption of water. Thus, it is easier to distinguish the spectral features of the macroplastics using the ratio curves. This is an advantage of the ratio curves over the use of the attenuation length ($kL$), the absorption ($\alpha$), and the extinction coefficients ($k$). The suppression of the spectral features comes from the anomalous dispersion that is strongest in the vicinity of local absorption maximum. The influence of the anomalous dispersion on the absorption maximum has been exploited to tune the absorption of dyes and the refractive index of water using K-K relations [28].

For certain (meso)plastics, the peaks observed around 1200 nm are subject to variations because of the different methods of production. The magnitude, trend, and the presence of peaks in the ratio curves enable differentiation among different plastic types. Although the K-K relation assumes non-scattering of photons, scattering distorts the obtained optical path length, $\Delta nL$. Similar distortion also occurs in the attenuation length, $kL$, but with an explicit dependence on $L$. In the case of both smooth and rough plastic, we can use the ratio curve to differentiate smooth from rough macroplastics of the same plastic and identify the type by the (dis)appearance or the shift of the peak positions in the spectral features of the ratio curve.

Once the plastic type has been identified using K-K analysis and the ratio curves, it is possible to use a library of extinction coefficient spectra for smooth (meso)plastics measured in the laboratory to calculate the thickness of the identified sample by using Beer-Lambert's law. In the case of a rough sample, Beer-Lambert's law will fail to result in thickness values much larger than the ones obtained for the smooth sample. This failure is due to the breakdown of Beer-Lambert's law for scattering medium, leading to the identification of rough samples.

## 5. Conclusions

In this paper, we have introduced a novel method for the screening and assessment of the surface quality of macroplastics of different plastic types with unknown thickness in water. Usually, the knowledge of sample thickness is mandatory for the identification and screening of objects using conventional methods. However, we exploit a non-conventional K-K relation and the spectral properties to detect these samples. The method is based on the calculation ratio (curves) of the attenuation length and the change in optical path length of incident radiation, which are obtained from the transmittance and the K-K relation, respectively. This ratio calculation allows the possibility to circumvent the limitation imposed by the sample thickness. Since the method is independent of sample thickness, and since the ratio function is detected from an area that corresponds to the size of a MP, it will be a useful technique for in situ identification of relatively large microplastics that can be probed with spectrophotometer.

**Author Contributions:** Conceptualization K.E.-P. and B.E.K.; Sample Preparation M.U.I., B.A., J.A., J.R.; Data Curation M.U.I., B.A., J.A., J.R.; Writing—Original draft preparation, B.E.K., K.E.-P. and B.A.; Supervision of the entire work, K.-E.P.; Writing-Review & Editing, B.E.K., K.-E.P. and B.A. All authors have read and agreed to the published version of the manuscript.

**Funding:** This research received no external funding.

**Acknowledgments:** This work is part of the Academy of Finland Flagship Programme, Photonics Research and Innovation (PREIN), decision 321066.

**Conflicts of Interest:** Authors declares no conflict of interest.

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
