# Peer review of "Identification of Plastic Type and Surface Roughness of Film-Type Plastics in Water Using Kramers–Kronig Analysis"

_chemosensors, doi:10.3390/chemosensors8040088_

Round 1
Reviewer 1 Report
Review of the manuscript entitled ‘On Identification of the Plastic-Type and Ascertaining the Nature of Plastic Surface Using Kramers-Kronig Analysis’
The goal of the manuscript is to find a method of detection of microplastics. For in-situ measurement conditions, both plastic-type and thickness of the film are unknown and an analytical method like Kramers-Kronig (K-K) is better suited for such a problem.
In this work, authors apply a modified conventional K-K relation on inverting transmittance for the identification of smooth and rough film type MPs with unknown thickness. The method is verified using the transmittance data of smooth and roughened film type microplastics. The key point is to exploit a library of ratio function obtained with the aid of the intrinsic optical properties of the plastics.
The work is interesting and deserves to be published.
However, the manuscript must be corrected and contains a lot of English spellings.
The ratio function obtained with the aid of 84 K-K analysis is used to characterize plastic types and to ascertain information on the surface 85 roughness of plastics in the aquatic environment.
Authors must describe the preparation of the samples. It is difficult for the reader to have a clear understanding of the nature of samples. Do you have the same sample sizes? What does it mean ‘rough’, smooth’?
Authors say that: ‘The curve for smooth plastic has a faint peak at 1161 nm and a stronger peak at 1211 nm, while the curve for rough plastic has a weaker peak at 1166 nm and a stronger peak at 1212 nm.’ However, I cannot see that in the figure 2a and b. Could you insert a figure in the figure a with a zoom?
Line 190: Authors say that: ‘the spectral position of the intersection points will change with changing surface roughness’. How do you observe that?
Line 241: what is the function ‘r’?
What is the sensitivity of the method to the roughness? For example in the figure 4d. How the roughness will affect the result?
Author Response
AUTHOR RESPONSE TO REVIEWER 1
Dear Reviewer
We thank you for your constructive comments which have really assisted us to increase and improve the scientific quality of our work. We have implemented changes in the manuscript in RED according to your suggestions and here are our responses.
- However, the manuscript must be corrected and contains a lot of English spellings.
Response
We apologize for English spelling errors the text has been edited to improve the clarity of the manuscript. We hope the reviewer finds the text more reader-friendly.
- Authors must describe the preparation of the samples. It is difficult for the reader to have a clear understanding of the nature of samples. Do you have the same sample sizes? What does it mean ‘rough’, smooth’?
Response
We agree with the reviewer on the matter. Our intention was to refer readers to the experimental section of [1] since we used similar data, but we failed to clearly communicate. However, to convey a complete message, we expanded the experimental section.
The meaning of the words ‘rough’ and ‘smooth’ is now given in the first place where they appear in the text. The film samples having both faces smooth (smooth) and film samples having both faces rough (rough).
In text- Page 5-7; lines 127 – 152:
The samples studied are commercial plastics: polypropylene (PP), polyethylene terephthalate (PET), polyamide nylon 6 (PA) polystyrene (PS), and low-density polyethylene (LDPE) which were purchased from Goodfellow, UK. Thin rectangular sheets with an approximate size of 55 mm x 7 mm were cut from the large pristine plastic samples to represent the smooth macroplastics. Next, a larger piece of the pristine sample was roughened on both sides using different sandpaper grits in a circular pattern. A similar size of the smooth macroplastic was then cut from the roughened larger piece. The roughening was performed to simulate the mechanical erosion of plastics in natural water bodies. Tables 1 and 2, respectively, show the thickness of the samples as purchased and the obtained average surface roughness, measured with a stylus profilometer (Mitutoyo SJ-210, Japan), for the different sandpaper grits after roughening. The thickness of PET plastic comparable to the thickness of plastic mineral water bottles. We, therefore, consider the special case of PET bottles having the same average surface roughness on both sides for different magnitudes to simulate of wear of plastics in water bodies.
The smooth and rough macroplastics were immersed in a 1mm-thick cuvette, containing tap water to form a complex interface (air-glass-water-plastic-water-glass-air), and was measured with the spectrophotometer (Perkin-Elmer Lambda 9) in the Vis-NIR spectral range in under standard laboratory conditions. The smaller cuvette was chosen to reduce the absorption in the measured spectral range. The recorded transmission spectra for the different samples are available in [1].
- The authors say that: ‘The curve for smooth plastic has a faint peek at 1161 nm and a stronger peak at 1211 nm, while the curve for rough plastic has a weaker peak at 1166 nm and a stronger peak at 1212 nm.’ However, I cannot see that in figure 2a and b. Could you insert a figure in the figure with a zoom?
Response
We believe that the reviewer is referring to Fig. 3a, which shows the ratio curves for the PS sample. The size of all the figures has been increased to enhance the visibility of the peaks.
- Line 190: Authors say that: ‘the spectral position of the intersection points will change with changing surface roughness’. How do you observe that?
In text—Page 9; Lines 185-188:
We note that the wavelength at which the smooth and rough macroplastics ratio curves intersect is likely to vary with changing average surface roughness. This conclusion is because, the local transmittance is expected to increase when the average surface roughness corresponds with the wavelength in accordance with Mie scattering.
- Line 241: what is the function ‘r’?
Response
The letter r was written by mistake, the correct letter is (σ) which denotes the ratio function. This has been corrected in the text.
- What is the sensitivity of the method to the roughness? For example, in the figure 4d. How the roughness will affect the result?
Response
The sensitivity of the method to the roughness is explained with the below text in red which has been added in two places within the manuscript. Moreover, the exponential behavior of transmittance versus roughness was observed in [27] at a fixed wavelength. This can be used to estimate the sensitivity of the present method versus average surface roughness, because in the first place prior to the Kramers-Kronig analysis, it is the transmittance that is measured. Sensitivity to find a change of at least 300 nm of average roughness is reasonable.
We remark that in Fig. 4 in [1] it is shown that the sensitivity of the transmittance against average surface roughness (at the spectral range starting from 500-1500 nm) is rather high because it is possible to screen between sample having average surface roughness magnitude 0.34 micrometers and a smooth plastic sample. Moreover, samples having a higher magnitude of surface microroughness could also be screened as shown in Fig. 4 of [1].
The ratio curves in Fig 5 show sensitivity to different average surface roughness resulting in an increase in the magnitude of ratio and or shift in peak position. Such sensitivity is useful for differentiating among different macroplastics, with surface roughness on either or both surfaces, as well as from water.
Reviewer 2 Report
In this manuscript (chemosensors-931465), Kanyathare et al. present a novel method for the determination of the thickness of a polymer film by examining the ratio between the real and imaginary part of the refractive index of the sample in the Vis/NIR spectral region. The claimed utility of the proposed approach should find use in monitoring microplastics pollution (in this case, in water).
The proposed method is original and interesting, and the significance of such aim for environment and health gives this study the necessary justification.
That being said, in its current state this manuscript suffers from two major issues that need to be properly addressed by the authors before this manuscript may be recommended for publication.
Firstly, this work lacks in clarity and several matters very important for the validity of this work are not entirely explained. Secondly, it remains unconvincing (for several reasons) that the results presented here are really meaningful for practical applications, which as authors claim, was the primary motivation for the study.
- It is unclear what was the thickness of the polymer films used in this study. Therefore, it remains unclear how these samples represent the target objects claimed to be microplastic films present as the pollutants in the environment. The ‘Experimental’ section lacks in detail.
- It is unclear how “size” in the “size between 1-5 mm” (‘Conclusions’ section) should be understood.
- "Appearance of such interference fringes would give information on the thickness of the plastic film." The appearance and the magnitude of the interference fringe strongly depends on the ratio between the refractive index of the polymer and the adjacent medium. While the polymer film in water may not be suitable for using the period of the interference fringe [because of n(water) being realtively high] to determine the thickness of the film, it may be possible if the film is surrounded by air. It is not clear, whether it is a practical necessity to examine the film in water.
- “"Unfortunately, the complex refractive index cannot be measured at optical frequencies whereas information to it can be measured". Besides rather awkward style of this sentence, this statement is not correct. Laser spectroscopy is capable of direct measurement of both components of the complex refractive index. A recent paper has demonstrated such capability in practical, analytical application (DOI: 10.1002/cphc.202000018)
- Please check, if the reference format present in the manuscript is the accepted one by Chemosensors.
Author Response
AUTHOR RESPONSE TO REVIEWER 2
Dear Reviewer
We thank you for your constructive comments which have assisted us to increase and improve the scientific quality of our work. We have implemented changes in the manuscript in RED according to your suggestions and here are our responses.
Reviewer report:
- It is unclear what was the thickness of the polymer films used in this study. Therefore, it remains unclear how these samples represent the target objects claimed to be microplastic films present as the pollutants in the environment. The ‘Experimental’ section lacks in detail.
Response
We agree with the reviewer on the matter. Our intention was to refer readers to the experimental section of [1] since we used similar data, but we failed to clearly communicate. However, to convey a complete message, we expanded the experimental section.
In text- Page 5-7; lines 127 – 152:
The samples studied are commercial plastics: polypropylene (PP), polyethylene terephthalate (PET), polyamide nylon 6 (PA) polystyrene (PS), and low-density polyethylene (LDPE) which were purchased from Goodfellow, UK. Thin rectangular sheets with an approximate size of 55 mm x 7 mm were cut from the large pristine plastic samples to represent the smooth macroplastics. Next, a larger piece of the pristine sample was roughened on both sides using different sandpaper grits in a circular pattern. A similar size of the smooth macroplastic was then cut from the roughened larger piece. The roughening was performed to simulate the mechanical erosion of plastics in natural water bodies. Tables 1 and 2, respectively, show the thickness of the samples as purchased and the obtained average surface roughness, measured with a stylus profilometer (Mitutoyo SJ-210, Japan), for the different sandpaper grits after roughening. The thickness of PET plastic comparable to the thickness of plastic mineral water bottles. We, therefore, consider the special case of PET bottles having the same average surface roughness on both sides for different magnitudes to simulate of wear of plastics in water bodies.
Table 1: Plastic materials and thickness as purchased.
|
Sample |
Thickness (mm) |
|
PP |
0.55 |
|
PET |
0.25 |
|
PA |
0.50 |
|
PS |
0.19 |
|
LDPE |
0.50 |
Table 2: Sandpaper grit size and the measured average surface roughness of the samples.
|
Samples |
Grit |
Average surface roughness Ra (μm) |
|
|
1200 |
0.34 ± 0.04 |
|
Rough |
320 |
1.10 ± 0.17 |
|
Pristine |
- |
< 0.10 |
The smooth and rough macroplastics were immersed in a 1mm-thick cuvette, containing tap water to form a complex interface (air-glass-water-plastic-water-glass-air), and was measured with the spectrophotometer (Perkin-Elmer Lambda 9) in the Vis-NIR spectral range in under standard laboratory conditions. The smaller cuvette was chosen to reduce the absorption in the measured spectral range. The recorded transmission spectra for the different samples are available in [1].
- It is unclear how “size” in the “size between 1-5 mm” (‘Conclusions’ section) should be understood.
Response
We apologize for the possible ambiguity in our statement. In the experimental section, the size of the samples is given as 55mm x 7mm. We have revised the conclusion to remove the ambiguity
In-text—Page 17; Lines 303-312:
In this paper, we have introduced a novel method for the screening and assessment of the surface quality of macroplastics of different plastic types with unknown thickness in water. Usually, the knowledge of sample thickness is mandatory for the identification and screening of objects using conventional methods. However, we exploit a modified form of the Kramers-Kronig (K-K) relation and the spectral properties to detect these samples. The method is based on the calculation ratio (curves) of the attenuation length and the change in optical path length of incident radiation, which is obtained from the transmittance and the K-K relation, respectively. This ratio calculation allows the possibility to circumvent the limitation imposed by the sample thickness. Since the method is independent of sample thickness, by extension, it will be a useful technique for in-situ identification of relatively large microplastics that can be probed with a spectrophotometer.
- The appearance of such interference fringes would give information on the thickness of the plastic film." The appearance and the magnitude of the interference fringe strongly depend on the ratio between the refractive index of the polymer and the adjacent medium. While the polymer film in water may not be suitable for using the period of the interference fringe [because of n(water) being relatively high] to determine the thickness of the film, it may be possible if the film is surrounded by air. It is not clear, whether it is a practical necessity to examine the film in water.
Response
During the inversion of transmission data using conventional K-K analysis usually thickness of the sample is required in the calculation, in that sense, it is a necessity especially when considering plastics in the water. On the contrary, in this work, we have applied a novel method of a modified K-K analysis that does not require a priori knowledge of the sample thickness.
- Unfortunately, the complex refractive index cannot be measured at optical frequencies whereas information to it can be measured". Besides the rather awkward style of this sentence, this statement is not correct. Laser spectroscopy is capable of direct measurement of both components of the complex refractive index. A recent paper has demonstrated such capability in the practical, analytical application (DOI: 10.1002/cphc.202000018).
Response
We agree with the reviewer and apologies for the misinformation. Indeed, it is the complex permittivity that cannot be directly measured at the optical frequency. Such has been removed.
- Please check if the reference format present in the manuscript is the accepted one by Chemosensors.
Response
The references have been corrected to match the standards of chemosensors, see the reference section.
Round 2
Reviewer 1 Report
The corrections done presented on the new manuscript version match the requirements indicated in the review. The manuscript deserves to be published.
Reviewer 2 Report
A careful evaluation of the revised manuscript indicated that the Authors adequately addressed the major concerns identified upon the review of the original manuscript. In its present version this manuscript is suitable for publication.